# AdaSpeech: Adaptive Text to Speech for Custom Voice

**Mingjian Chen**[∗]**, Xu Tan,**[∗]**Bohan Li, Yanqing Liu, Tao Qin, Sheng Zhao, Tie-Yan Liu**
Microsoft Research Asia, Microsoft Azure Speech
`{xuta,taoqin,szhao,tyliu}@microsoft.com`

## Abstract

Custom voice, a specific text to speech (TTS) service in commercial speech platforms, aims to adapt a source TTS model to synthesize personal voice for a target speaker using few speech from her/him. Custom voice presents two unique challenges for TTS adaptation: 1) to support diverse customers, the adaptation model needs to handle diverse acoustic conditions which could be very different from source speech data, and 2) to support a large number of customers, the adaptation parameters need to be small enough for each target speaker to reduce memory usage while maintaining high voice quality. In this work, we propose AdaSpeech, an adaptive TTS system for high-quality and efficient customization of new voices. We design several techniques in AdaSpeech to address the two challenges in custom voice: 1) To handle different acoustic conditions, we model the acoustic information in both utterance and phoneme level. Specifically, we use one acoustic encoder to extract an utterance-level vector and another one to extract a sequence of phoneme-level vectors from the target speech during pre-training and fine-tuning; in inference, we extract the utterance-level vector from a reference speech and use an acoustic predictor to predict the phoneme-level vectors. 2) To better trade off the adaptation parameters and voice quality, we introduce conditional layer normalization in the mel-spectrogram decoder of AdaSpeech, and fine-tune this part in addition to speaker embedding for adaptation. We pre-train the source TTS model on LibriTTS datasets and fine-tune it on VCTK and LJSpeech datasets (with different acoustic conditions from LibriTTS) with few adaptation data, e.g., 20 sentences, about 1 minute speech. Experiment results show that AdaSpeech achieves much better adaptation quality than baseline methods, with only about 5K specific parameters for each speaker, which demonstrates its effectiveness for custom voice. The audio samples are available at `https://speechresearch.github.io/adaspeech/`.

## 1 Introduction

Text to speech (TTS) aims to synthesize natural and intelligible voice from text, and attracts a lot of interests in machine learning community (Arik et al., 2017; Wang et al., 2017; Gibiansky et al., 2017; Ping et al., 2018; Shen et al., 2018; Ren et al., 2019). TTS models can synthesize natural human voice when training with a large amount of high-quality and single-speaker recordings (Ito, 2017), and has been extended to multi-speaker scenarios (Gibiansky et al., 2017; Ping et al., 2018; Zen et al., 2019; Chen et al., 2020) using multi-speaker corpora (Panayotov et al., 2015; Veaux et al., 2016; Zen et al., 2019). However, these corpora contain a fixed set of speakers where each speaker still has a certain amount of speech data.

Nowadays, custom voice has attracted increasing interests in different application scenarios such as personal assistant, news broadcast and audio navigation, and has been widely supported in commercial speech platforms (some custom voice services include Microsoft Azure, Amazon AWS and Google Cloud). In custom voice, a source TTS model is usually adapted on personalized voices with few adaptation data, since the users of custom voice prefer to record as few adaptation data as possible (several minutes or seconds) for convenient purpose. Few adaptation data presents great challenges

---

[∗]The first two authors contribute equally to this work. Corresponding author: Xu Tan, xuta@microsoft.com.

on the naturalness and similarity of adapted voice. Furthermore, there are also several distinctive challenges in custom voice: 1) The recordings of the custom users are usually of different acoustic conditions from the source speech data (the data to train the source TTS model). For example, the adaptation data is usually recorded with diverse speaking prosodies, styles, emotions, accents and recording environments. The mismatch in these acoustic conditions makes the source model difficult to generalize and leads to poor adaptation quality. 2) When adapting the source TTS model to a new voice, there is a trade-off between the fine-tuning parameters and voice quality. Generally speaking, more adaptation parameters will usually result in better voice quality, which, as a result, increases the memory storage and serving cost[1].

While previous works in TTS adaptation have well considered the few adaptation data setting in custom voice, they have not fully addressed the above challenges. They fine-tune the whole model (Chen et al., 2018; Kons et al., 2019) or decoder part (Moss et al., 2020; Zhang et al., 2020), achieving good quality but causing too many adaptation parameters. Reducing the amount of adaptation parameters is necessary for the deployment of commercialized custom voice. Otherwise, the memory storage would explode as the increase of users. Some works only fine-tune the speaker embedding (Arik et al., 2018; Chen et al., 2018), or train a speaker encoder module (Arik et al., 2018; Jia et al., 2018; Cooper et al., 2020; Li et al., 2017; Wan et al., 2018) that does not need fine-tuning during adaptation. While these approaches lead a light-weight and efficient adaptation, they result in poor adaptation quality. Moreover, most previous works assume the source speech data and adaptation data are in the same domain and do not consider the setting with different acoustic conditions, which is not practical in custom voice scenarios.

In this paper, we propose AdaSpeech, an adaptive TTS model for high-quality and efficient customization of new voice. AdaSpeech employ a three-stage pipeline for custom voice: 1) pre-training; 2) fine-tuning; 3) inference. During the pre-training stage, the TTS model is trained on large-scale multi-speaker datasets, which can ensure the TTS model to cover diverse text and speaking voices that is helpful for adaptation. During the fine-tuning stage, the source TTS model is adapted on a new voice by fine-tuning (a part of) the model parameters on the limited adaptation data with diverse acoustic conditions. During the inference stage, both the unadapted part (parameters shared by all custom voices) and the adapted part (each custom voice has specific adapted parameters) of the TTS model are used for the inference request. We build AdaSpeech based on the popular non-autoregressive TTS models (Ren et al., 2019; Peng et al., 2020; Kim et al., 2020; Ren et al., 2021) and further design several techniques to address the challenges in custom voice:

- Acoustic condition modeling. In order to handle different acoustic conditions for adaptation, we model the acoustic conditions in both utterance and phoneme level in pre-training and fine-tuning. Specifically, we use two acoustic encoders to extract an utterance-level vector and a sequence of phoneme-level vectors from the target speech, which are taken as the input of the mel-spectrogram decoder to represent the global and local acoustic conditions respectively. In this way, the decoder can predict speech in different acoustic conditions based on these acoustic information. Otherwise, the model would memorize the acoustic conditions and cannot generalize well. In inference, we extract the utterance-level vector from a reference speech and use another acoustic predictor that is built upon the phoneme encoder to predict the phoneme-level vectors.

- Conditional layer normalization. To fine-tune as small amount of parameters as possible while ensuring the adaptation quality, we modify the layer normalization (Ba et al., 2016) in the mel-spectrogram decoder in pre-training, by using speaker embedding as the conditional information to generate the scale and bias vector in layer normalization. In fine-tuning, we only adapt the parameters related to the conditional layer normalization. In this way, we can greatly reduce adaptation parameters and thus memory storage[2] compared with fine-tuning the whole model, but maintain high-quality adaptation voice thanks to the flexibility of conditional layer normalization.

To evaluate the effectiveness of our proposed AdaSpeech for custom voice, we conduct experiments to train the TTS model on LibriTTS datasets and adapt the model on VCTK and LJSpeech datasets with different adaptation settings. Experiment results show that AdaSpeech achieves better adaptation quality in terms of MOS (mean opinion score) and SMOS (similarity MOS) than baseline methods, with

---

[1]For example, to support one million users in a cloud speech service, if each custom voice consumes 100MB model sizes, the total memory storage would be about 100PB, which is quite a big serving cost.

[2]We further reduce the memory usage in inference as described in Section 2.3.

only about 5K specific parameters for each speaker, demonstrating its effectiveness for custom voice. Audio samples are available at `https://speechresearch.github.io/adaspeech/`.

## 2 ADASPEECH

In this section, we first describe the overall design of our proposed AdaSpeech, and then introduce the key techniques to address the challenges in custom voice. At last, we list the pre-training, fine-tuning and inference pipeline of AdaSpeech for custom voice.

The model structure of AdaSpeech is shown in Figure 1. We adopt FastSpeech 2 (Ren et al., 2021) as the model backbone considering the FastSpeech (Ren et al., 2019; 2021) series are one of the most popular models in non-autoregressive TTS. The basic model backbone consists of a phoneme encoder, a mel-spectrogram decoder, and a variance adaptor which provides variance information including duration, pitch and energy into the phoneme hidden sequence following Ren et al. (2021). As shown in Figure 1, we design two additional components to address the distinctive challenges in custom voice: 1) to support diverse customers, we use acoustic condition modeling to capture the diverse acoustic conditions of adaptation speech in different granularities; 2) to support a large number of customers with affordable memory storage, we use conditional layer normalization in decoder for efficient adaptation with few parameters while high voice quality. In the next subsections, we introduce the details of these components respectively.

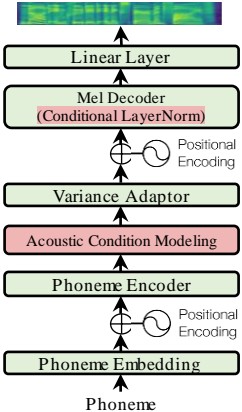

Figure 1: AdaSpeech.

### 2.1 ACOUSTIC CONDITION MODELING

In custom voice, the adaptation data can be spoken with diverse prosodies, styles, accents, and can be recorded under various environments, which can make the acoustic conditions far different from that in source speech data. This presents great challenges to adapt the source TTS model, since the source speech cannot cover all the acoustic conditions in custom voice. A practical way to alleviate this issue is to improve the adaptability (generalizability) of source TTS model. In text to speech, since the input text lacks enough acoustic conditions (such as speaker timbre, prosody and recording environments) to predict the target speech, the model tends to memorize and overfit on the training data (Ren et al., 2021), and has poor generalization during adaptation. A natural way to solve such problem is to provide corresponding acoustic conditions as input to make the model learn reasonable text-to-speech mapping towards better generalization instead of memorizing.

To better model the acoustic conditions with different granularities, we categorize the acoustic conditions in different levels as shown in Figure 2a: 1) speaker level, the coarse-grained acoustic conditions to capture the overall characteristics of a speaker; 2) utterance level, the fine-grained acoustic conditions in each utterance of a speaker; 3) phoneme level, the more fine-grained acoustic conditions in each phoneme of an utterance, such as accents on specific phonemes, pitches, prosodies and temporal environment noises[3]. Since speaker ID (embedding) is widely used to capture speaker-level acoustic conditions in multi-speaker scenario (Chen et al., 2020), speaker embedding is used by default. We describe the utterance-level and phoneme-level acoustic condition modeling as follows.

- Utterance Level. We use an acoustic encoder to extract a vector from a reference speech, similar to Arik et al. (2018); Jia et al. (2018); Cooper et al. (2020), and then expand and add it to the phoneme hidden sequence to provide the utterance-level acoustic conditions. As shown in Figure 2b, the acoustic encoder consists of several convolutional layers and a mean pooling layer to get a single vector. The reference speech is the target speech during training, while a randomly chosen speech of this speaker during inference.

- Phoneme Level. We use another acoustic encoder (shown in Figure 2c) to extract a sequence of phoneme-level vectors from the target speech and add it to the phoneme hidden sequence to

---

[3]Generally, more fine-grained frame-level acoustic conditions (Zhang et al., 2021) exist, but have marginal benefits considering their prediction difficulty. Similarly, more coarse-grained language level conditions also exist, but we do not consider multilingual setting in this work and leave it for future work.

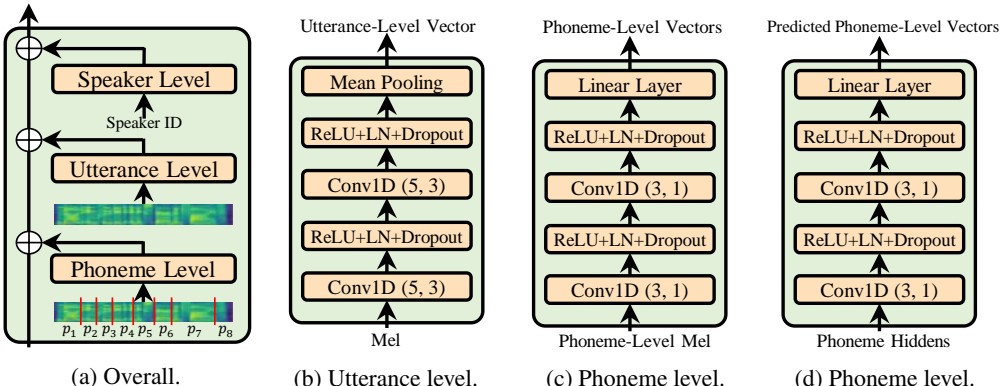

| (a) Overall. | (b) Utterance level. | (c) Phoneme level. | (d) Phoneme level. |

Figure 2: (a) The overall structure of acoustic condition modeling. (b) Utterance-level acoustic encoder. (c) Phoneme-level acoustic encoder, where phoneme-level mel means the mel-frames aligned to the same phoneme are averaged. (d) Phoneme-level acoustic predictor, where phoneme hiddens is the hidden sequence from the phoneme encoder in Figure 1. 'Conv1D $(m, n)$' means the kernel size and stride size in 1D convolution is $m$ and $n$ respectively. 'LN' means layer normalization. As shown in Figure 2a, the phoneme-level vectors are directly added element-wisely into the hidden sequence, and the utterance-level and speaker level vector/embedding are first expanded to the same length and then added element-wisely into the hidden sequence.

provide the phoneme-level acoustic conditions[4]. In order to extract phoneme-level information from speech, we first average the speech frames corresponding to the same phoneme according to alignment between phoneme and mel-spectrogram sequence (shown in Figure 2a), to convert to length of speech frame sequence into the length of phoneme sequence, similar to Sun et al. (2020); Zeng et al. (2020). During inference, we use another phoneme-level acoustic predictor (shown in Figure 2d) which is built upon the original phoneme encoder to predict the phoneme-level vectors.

Using speech encoders to extract a single vector or a sequence of vectors to represent the characteristics of a speech sequence has been adopted in previous works (Arik et al., 2018; Jia et al., 2018; Cooper et al., 2020; Sun et al., 2020; Zeng et al., 2020). They usually leverage them to improve the speaker timbre or prosody of the TTS model, or improve the controllability of the model. The key contribution in our acoustic condition modeling in this work is the novel perspective to model the diverse acoustic conditions in different granularities to make the source model more adaptable to different adaptation data. As analyzed in Section 4.2, utterance-level and phoneme-level acoustic modeling can indeed help the learning of acoustic conditions and is critical to ensure the adaptation quality.

## 2.2 CONDITIONAL LAYER NORMALIZATION

Achieving high adaptation quality while using small adaptation parameters is challenging. Previous works use zero-shot adaptation with speaker encoder (Arik et al., 2018; Jia et al., 2018; Cooper et al., 2020) or only fine-tune the speaker embedding cannot achieve satisfied quality. Can we greatly increase the voice quality at the cost of slightly more but negligible parameters? To this end, we analyze the model parameters of FastSpeech 2 (Ren et al., 2021), which is basically built upon

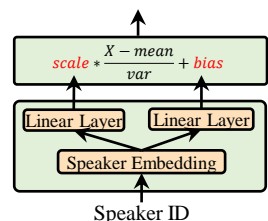

Figure 3: Conditional LayerNorm.

the structure of Transformer (Vaswani et al., 2017), with a self-attention network and a feed-forward network in each Transformer block. Both the matrix multiplications in the query, key, value and output of self-attention and two-layer feed-forward networks are parameter-intensive, which is not efficient to adapt. We find that layer normalization (Ba et al., 2016) is adopted in each self-attention and feed-forward network in decoder, which can greatly influence the hidden activation and final prediction with a light-weight learnable scale vector $\gamma$ and bias vector $\beta$: $LN(x) = \gamma \frac{x-\mu}{\sigma} + \beta$, where $\mu$ and $\sigma$ are the mean and variance of hidden vector $x$.

---

[4]Note that although the extracted vectors can contain all phoneme-level acoustic conditions ideally, we still use pitch and energy in the variance adaptor (shown in Figure 1) as additional input following Ren et al. (2021), in order to ease the burden of acoustic condition learning and focus on learning other acoustic conditions. We also tried to remove pitch and energy but found it causes worse adaptation quality.

If we can determine the scale and bias vector in layer normalization with the corresponding speaker characteristics using a small conditional network, then we can fine-tune this conditional network when adapting to a new voice, and greatly reduce the adaptation parameters while ensuring the adaptation quality. As shown in Figure 3, the conditional network consists of two simple linear layers $W_c^\gamma$ and $W_c^\beta$ that take speaker embedding $E^s$ as input and output the scale and bias vector respectively:

$$\gamma_c^s = E^s * W_c^\gamma, \ \ \beta_c^s = E^s * W_c^\beta, \tag{1}$$

where $s$ denotes the speaker ID, and $c \in [C]$ denotes there are $C$ conditional layer normalizations in the decoder (the number of decoder layer is $(C-1)/2$ since each layer has two conditional layer normalizations corresponding to self-attention and feed-forward network in Transformer, and there is an additional layer normalization at the final output) and each uses different conditional matrices.

## 2.3 Pipeline of AdaSpeech

We list the pre-training, fine-tuning and inference pipeline of AdaSpeech in Algorithm 1. During fine-tuning, we only fine-tune the two matrices $W_c^\gamma$ and $W_c^\beta$ in each conditional layer normalization in decoder and the speaker embedding $E^s$, fixing other model parameters including the utterance-level and phoneme-level acoustic encoders and phoneme-level acoustic predictor as described in Section 2.1. During inference, we do not directly use the two matrices $W_c^\gamma$ and $W_c^\beta$ in each conditional layer normalization since they still have large parameters. Instead we use the two matrices to calculate each scale and bias vector $\gamma_c^s$ and $\beta_c^s$ from speaker embedding $E_s$ according to Equation 1 considering $E_s$ is fixed in inference. In this way, we can save a lot of memory storage[5].

---

**Algorithm 1** Pre-training, fine-tuning and inference of AdaSpeech

1: **Pre-training**: Train the AdaSpeech model $\theta$ with source training data $D$.
2: **Fine-tuning**: Fine-tune $W_c^\gamma$ and $W_c^\beta$ in each conditional layer normalization $c \in [C]$ and speaker embedding $E^s$ with the adaptation data $D^s$ for each custom speaker/voice $s$.
3: **Inference**: *Deployment*: 1) Calculate $\gamma_c^s, \beta_c^s$ in each conditional layer normalization $c \in [C]$, and get the parameters $\theta^s = \{\{\gamma_c^s, \beta_c^s\}_{c=1}^C, E^s\}$ for speaker $s$. 2) Deploy the shared model parameters $\tilde{\theta}$ (not fine-tuned in $\theta$ during adaptation) and speaker specific parameters $\theta^s$ for $s$. *Inference*: Use $\tilde{\theta}$ and $\theta^s$ to synthesize custom voice for speaker $s$.

---

## 3 Experimental Setup

**Datasets**  We train the AdaSpeech source model on LibriTTS (Zen et al., 2019) dataset, which is a multi-speaker corpus (2456 speakers) derived from LibriSpeech (Panayotov et al., 2015) and contains 586 hours speech data. In order to evaluate AdaSpeech in custom voice scenario, we adapt the source model to the voices in other datasets including VCTK (Veaux et al., 2016) (a multi-speaker datasets with 108 speakers and 44 hours speech data) and LJSpeech (Ito, 2017) (a single-speaker high-quality dataset with 24 hours speech data), which have different acoustic conditions from LibriTTS. As a comparison, we also adapt the source model to the voices in the same LibriTTS dataset.

We randomly choose several speakers (including both male and female) from the training set of LibriTTS and VCTK and the only single speaker from the training set of LJSpeech for adaptation. For each chosen speaker, we randomly choose $K = 20$ sentences for adaptation and also study the effects of smaller $K$ in experiment part. We use all the speakers in the training set of LibriTTS (exclude those chosen for adaptation) to train the source AdaSpeech model, and use the original test sets in these datasets corresponding to the adaptation speakers to evaluate the adaptation voice quality.

We conduct the following preprocessing on the speech and text data in these corpora: 1) convert the sampling rate of all speech data to 16kHz; 2) extract the mel-spectrogram with 12.5ms hop size and 50ms window size following the common practice in Shen et al. (2018); Ren et al. (2019); 3) convert

---

[5]Assume the dimension of speaker embedding and hidden vector are both $h$, the number of conditional layer normalization is $C$. Therefore, the number of adaptation parameters are $2h^2C + h$, where the first 2 represents the two matrices for scale and bias vectors, and the second term $h$ represents the speaker embedding. If $h = 256$ and $C = 9$, the total number of parameters are about 1.2M, which is much smaller compared the whole model (31M). During deployment for each custom voice, the total additional model parameters for a new voice that need to be stored in memory becomes $2hC + h$, which is extremely small (4.9K in the above example).

text sequence into phoneme sequence with grapheme-to-phoneme conversion (Sun et al., 2019) and take phoneme as the encoder input.

**Model Configurations** The model of AdaSpeech follows the basic structure in FastSpeech 2 (Ren et al., 2021), which consists of 4 feed-forward Transformer blocks for the phoneme encoder and mel-spectrogram decoder. The hidden dimension (including the phoneme embedding, speaker embedding, the hidden in self-attention, and the input and output hidden of feed-forward network) is set to 256. The number of attention heads, the feed-forward filter size and kernel size are set to 2, 1024 and 9 respectively. The output linear layer converts the 256-dimensional hidden into 80-dimensional mel-spectrogram. Other model configurations follow Ren et al. (2021) unless otherwise stated.

The phoneme-level acoustic encoder (Figure 2c) and predictor (Figure 2d) share the same structure, which consists of 2 convolutional layers with filter size and kernel size of 256 and 3 respectively, and a linear layer to compress the hidden to a dimension of 4 (we choose the dimension of 4 according to our preliminary study and is also consistent with previous works (Sun et al., 2020; Zeng et al., 2020)). We use MFA (McAuliffe et al., 2017) to extract the alignment between the phoneme and mel-spectrogram sequence, which is used to prepare the input of the phoneme-level acoustic encoder. We also tried to leverage VQ-VAE (Sun et al., 2020) into the phoneme-level acoustic encoder but found no obvious gains. The utterance-level acoustic encoder consists of 2 convolutional layers with filter size, kernel size and stride size of 256, 5 and 3, and a pooling layer to obtain a single vector.

**Training, Adaptation and Inference** In the source model training process, we first train AdaSpeech for 60,000 steps, and all the model parameters are optimized except the parameters of phoneme-level acoustic predictor. Then we train AdaSpeech and the phoneme-level acoustic predictor jointly for the remaining 40,000 steps, where the output hidden of the phoneme-level acoustic encoder is used as the label (the gradient is stopped to prevent flowing back to the phoneme-level acoustic encoder) to train the phoneme-level acoustic predictor with mean square error (MSE) loss. We train AdaSpeech on 4 NVIDIA P40 GPUs and each GPU has a batch size of about 12,500 speech frames. Adam optimizer is used with $\beta_1 = 0.9$, $\beta_2 = 0.98$, $\epsilon = 10^{-9}$.

In the adaptation process, we fine-tune AdaSpeech on 1 NVIDIA P40 GPU for 2000 steps, where only the parameters of speaker embedding and conditional layer-normalization are optimized. In the inference process, the utterance-level acoustic conditions are extracted from another reference speech of the speaker, and the phoneme-level acoustic conditions are predicted from phoneme-level acoustic predictor. We use MelGAN (Kumar et al., 2019) as the vocoder to synthesize waveform from the generated mel-spectrogram.

## 4 RESULTS

In this section, we first evaluate the quality of the adaptation voices of AdaSpeech, and conduct ablation study to verify the effectiveness of each component in AdaSpeech, and finally we show some analyses of our method.

### 4.1 THE QUALITY OF ADAPTATION VOICE

We evaluate the quality of adaption voices in terms of naturalness (how the synthesized voices sound natural like human) and similarity (how the synthesized voices sound similar to this speaker). Therefore, we conduct human evaluations with MOS (mean opinion score) for naturalness and SMOS (similarity MOS) for similarity. Each sentence is listened by 20 judgers. For VCTK and LibriTTS, we average the MOS and SMOS scores of multiple adapted speakers as the final scores. We compare AdaSpeech with several settings: 1) GT, the ground-truth recordings; 2) GT mel + Vocoder, using ground-truth mel-spectrogram to synthesize waveform with MelGAN vocoder; 3) Baseline (spk emb), a baseline system based on FastSpeech2 which only fine-tunes the speaker embedding during adaptation, and can be regarded as our lower bound; 4) Baseline (decoder), another baseline system based on FastSpeech2 which fine-tunes the whole decoder during adaptation, and can be regarded as a strong comparable system since it uses more parameters during adaptation; 5) AdaSpeech, our proposed AdaSpeech system with utterance-/phoneme-level acoustic condition modeling and conditional layer normalization during adaptation[6].

---

[6]The audio samples are available at `https://speechresearch.github.io/adaspeech/`

| Metric | Setting | # Params/Speaker | LJSpeech | VCTK | LibriTTS |
|---|---|---|---|---|---|
| MOS | *GT* | / | $3.98 \pm 0.12$ | $3.87 \pm 0.11$ | $3.72 \pm 0.12$ |
| | *GT mel + Vocoder* | / | $3.75 \pm 0.10$ | $3.74 \pm 0.11$ | $3.65 \pm 0.12$ |
| | *Baseline (spk emb)* | 256 (256) | $2.37 \pm 0.14$ | $2.36 \pm 0.10$ | $3.02 \pm 0.13$ |
| | *Baseline (decoder)* | 14.1M (14.1M) | $3.44 \pm 0.13$ | $3.35 \pm 0.12$ | $3.51 \pm 0.11$ |
| | *AdaSpeech* | 1.2M (4.9K) | $3.45 \pm 0.11$ | $3.39 \pm 0.10$ | $3.55 \pm 0.12$ |
| SMOS | *GT* | / | $4.36 \pm 0.11$ | $4.44 \pm 0.10$ | $4.31 \pm 0.07$ |
| | *GT mel + Vocoder* | / | $4.29 \pm 0.11$ | $4.36 \pm 0.11$ | $4.31 \pm 0.07$ |
| | *Baseline (spk emb)* | 256 (256) | $2.79 \pm 0.19$ | $3.34 \pm 0.19$ | $4.00 \pm 0.12$ |
| | *Baseline (decoder)* | 14.1M (14.1M) | $3.57 \pm 0.12$ | $3.90 \pm 0.12$ | $4.10 \pm 0.10$ |
| | *AdaSpeech* | 1.2M (4.9K) | $3.59 \pm 0.15$ | $3.96 \pm 0.15$ | $4.13 \pm 0.09$ |

Table 1: The MOS and SMOS scores with $95\%$ confidence intervals when adapting the source AdaSpeech model (trained on LibriTTS) to LJSpeech, VCTK and LibriTTS datasets. The third column shows the number of additional parameters for each custom voice during adaptation (the number in bracket shows the number of parameters in inference following the practice in Section 2.3).

The MOS and SMOS results are shown in Table 1. We have several observations: 1) Adapting the model (trained on LibriTTS) to the cross-domain datasets (LJSpeech and VCTK) is more difficult than adapting to the in-domain datasets (LibriTTS), since the MOS and SMOS gap between the adaptation models (two baselines and AdaSpeech) and the ground-truth mel + vocoder setting is bigger on cross-domain datasets[7]. This also confirms the challenges of modeling different acoustic conditions in custom voice scenarios. 2) Compared with only fine-tuning speaker embedding, i.e., *Baseline (spk emb)*, AdaSpeech achieves significant improvements in terms of both MOS and SMOS in the three adaptation datasets, by only leveraging slightly more parameters in conditional layer normalization. We also analyze in next subsection (Table 3) that even if we increase the adaptation parameters of baseline to match or surpass that in AdaSpeech, it still performs much worse than AdaSpeech. 3) Compared with fine-tuning the whole decoder, i.e., *Baseline (decoder)*, AdaSpeech achieves slightly better quality in both MOS and SMOS and importantly with much smaller adaptation parameters, which demonstrates the effectiveness and efficiency of our proposed acoustic condition modeling and conditional layer normalization. Note that fine-tuning the whole decoder causes too much adaptation parameters that cannot satisfy the custom voice scenario.

## 4.2 METHOD ANALYSIS

In this section, we first conduct ablation studies to verify the effectiveness of each component in AdaSpeech, including utterance-level and phoneme-level acoustic condition modeling, and conditional layer normalization, and then conduct more detailed analyses on our proposed AdaSpeech.

| Setting | CMOS |
|---|---|
| *AdaSpeech* | 0 |
| *AdaSpeech w/o UL-ACM* | $-0.12$ |
| *AdaSpeech w/o PL-ACM* | $-0.21$ |
| *AdaSpeech w/o CLN* | $-0.14$ |

Table 2: The CMOS of the ablation study on VCTK. UL-ACM and PL-ACM represents utterance-level and phoneme-level acoustic condition modeling, and CLN represents conditional layer normalization.

**Ablation Study**    We compare the CMOS (comparison MOS) of the adaptation voice quality when removing each component in AdaSpeech on VCTK testset (each sentence is listened by 20 judgers). Specifically, when removing conditional layer normalization, we only fine-tune the speaker embedding. From Table 2, we can see that removing utterance-level and phoneme-level acoustic modeling, and conditional layer normalization all result in performance drop in voice quality, demonstrating the effectiveness of each component in AdaSpeech.

**Analyses on Acoustic Condition Modeling**    We analyze the vectors extracted from the utterance-level acoustic encoder for several speakers on LibriTTS datasets. We use t-SNE (Maaten & Hinton,

---

[7]For example, the MOS gaps of the three settings (two baselines and AdaSpeech) on LJSpeech are 1.38, 0.31, 0.30, and on VCTK are 1.38, 0.39, 0.35, respectively, which are bigger than that on LibriTTS (0.63, 0.14, 0.10).

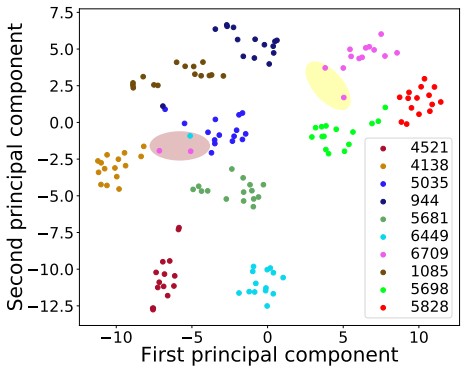

(a) Utterance-level visualization.

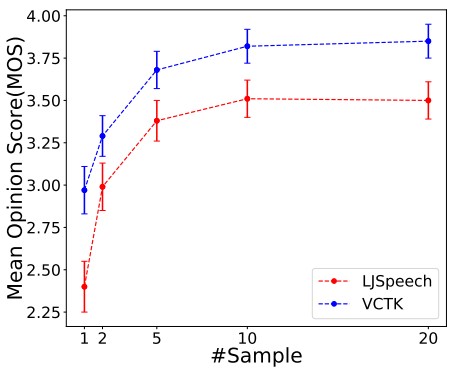

(b) MOS with varying data.

Figure 4: (a) The visualization of utterance-level acoustic vectors for several speakers (each number in the legend represents a speaker ID in LibriTTS datasets). (b) The MOS of different adaptation data on LJSpeech and VCTK.

2008) to illustrate them in Figure 4a, where each point represents an utterance-level vector and each color belongs to the same speaker. It can be seen that different utterances of the same speaker are clustered together but have difference in acoustic conditions. There are some exceptions, such as the two pink points one blue point in the brown solid circle. According to our investigation on the corresponding speech data, these points correspond to the utterances with short and emotional voice, and thus are close to each other although belonging to different speakers.

**Analyses on Conditional Layer Normalization**  We further compare conditional layer normalization (CLN) with other two settings: 1) LN + fine-tune scale/bias: removing the condition on speaker embedding, and only fine-tuning scale/bias in layer normalization and speaker embedding; 2) LN + fine-tuning others: removing the condition on speaker embedding, and instead fine-tuning other (similar or even larger amount of) parameters in the decoder[8]. The CMOS evaluations are shown in Table 3. It can be seen that both settings result in worse quality compared with conditional layer normalization, which verifies its effectiveness.

| Setting | CMOS |
|---|---|
| CLN | 0 |
| LN + fine-tune scale/bias | $-0.18$ |
| LN + fine-tune others | $-0.24$ |

Table 3: The CMOS on VCTK for the comparison of conditional layer normalization.

**Varying Adaptation Data**  We study the voice quality with different amount of adaptation data (fewer than the default setting) on VCTK and LJSpeech, and conduct MOS evaluation as shown in Figure 4b. It can be seen that the voice quality continue drops when adaptation data decreases, and drops quickly when the adaptation data is fewer than 10 sentences.

## 5  CONCLUSIONS

In this paper, we have developed AdaSpeech, an adaptive TTS system to support the distinctive requirements in custom voice. We propose acoustic condition modeling to make the source TTS model more adaptable for custom voice with various acoustic conditions. We further design conditional layer normalization to improve the adaptation efficiency: fine-tuning few model parameters to achieve high voice quality. We finally present the pipeline of pre-training, fine-tuning and inference in AdaSpeech for custom voice. Experiment results demonstrate that AdaSpeech can support custom voice with different acoustic conditions with few memory storage and at the same time with high voice quality. For future work, we will further improve the modeling of acoustic conditions in the source TTS model and study more diverse acoustic conditions such as noisy speech in custom voice. We will also investigate the adaptation setting with untranscribed data (Yan et al., 2021) and further compress the model size (Luo et al., 2021) to support more custom voices.

---

[8] According to the preliminary study, we found fine-tuning the last linear layer and the last feed-forward network in decoder can result in better performance than fine-tuning other part in decoder.

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
