# OpenReview forum: "AdaSpeech: Adaptive Text to Speech for Custom Voice"
_ICLR.cc/2021/Conference — ICLR 2021 Poster_

### Official Review · AnonReviewer4 · 2020-10-24
**AdaSpeech: Adaptive Text to Speech for Custom Voice**

**Rating:** 7
**Confidence:** 5

**Review:**

The authors propose an interesting text-to-speech adaptation method for high quality and efficient customization of new voice. The proposed method consists of two-stage modeling : multi-phonetic-level acoustic condition modeling and conditional layer normalization. In the first stage modeling, the authors proposed a new phoneme-level acoustic condition modeling in addition to the speaker and utterance-level approaches. In the second stage modeling, they employ conditional layer normalization for efficient adaptation.

Overall, I vote for ACCEPTING. The idea of TTS adaptation for customization of new voice is appealing to me. The proposed approach seems new and technically decent. Their experimental results are good and meaningful. The overall structure of the paper is systematic and well-written despite its minor errors and contains plenty of solid experimental results and discussion.

Pros
- This paper takes one important issue of current speech synthesis area: TTS adaptation to new voice.
- Its multi-phonetic-level acoustic condition modeling approach seem technically new and interesting,
- Its comprehensive experimental results well showed the effectiveness of the proposed approach.

Cons

This paper still has some issues that are conceptually not so convincing, which need to be clarified in the rebuttal session.

- In paper, it is said that the phoneme-level acoustic encoder uses phoneme-level Mel features as its input. In the inference, the phoneme-level acoustic predictor uses phoneme hiddens as its input to predict phoneme-level vectors. I think the Mel features used in phoneme-level acoustic encoder contain personal voice information. On the contrary, the phoneme hiddens used in phoneme-level acoustic predictor do not seem to contain any personal voice information because they are resulted from the phoneme encoder that uses text information only as its input in Fig 1.  Please clarify this issue.
- The overall structure of acoustic condition modeling in Figure 2 (a) is not so clear. For higher reproducibility, authors need to describe it in more detail. The addition of three, that is, speaker, utterance, and phoneme level vectors to the output of phoneme encoder can be done either in element-by-element or by concatenation.
- In section 2.3, authors described that during inference, they do not use the two matrices, but it seems difficult to understand how gamma and beta variables could be calculated using eq. 1 without them?

Some typos:

- In pages 2 and 7, describe full representation of MOS, SMOS, and CMOS, respectively.
- In page 3, random chosen  -->  randomly chosen
- In pages 2, 4, while ensure -->  while ensuring

---

> ### Author Response · Authors · 2020-11-19
> **Response to AnonReviewer4**
>
> Thanks for your positive comments. We reply to your questions and comments as follows.
>
> **[How the phoneme-level acoustic predictor can predict the acoustic information beyond prosody such as personal voice information purely based on phoneme sequence? Please clarify this issue. ]**
>
> We clarify this problem with the following points:
> 1. We let the phoneme- and utterance-level acoustic encoders to extract necessary information from different granularities to predict the correct mel-spectrogram, instead of overfitting due to lack of enough acoustic information.
> 2. The utterance-level acoustic encoder extracts a single vector to summarize the whole utterance, which tends to model general personal voice information. Our experiments also show that utterance-level vector can model personal voice information.
> 3. The phoneme-level acoustic encoder extracts a sequence of low-dimension vectors (the dimension is only 4 as shown in the Model Configurations part in Section 3 of the paper) from the averaged speech sequence (averaged from frame level into phoneme level), which tends to model fine-grained information like prosody, instead of personal voice information.
> 4. The low hidden dimension of the phoneme-level vectors is very important to ensure only phoneme-level information like prosody is extracted, instead of general personal voice information. We conducted preliminary studies on the dimension of the phoneme-level vectors and found that larger dimensions result in worse adaptation quality.
> 5. Therefore, in inference, the phoneme-level acoustic predictor can predict information like prosody based on phoneme encoder.
>
> **[The overall structure of acoustic condition modeling in Figure 2 (a) is not so clear.]**
>
> The three are added together by element-wise addition. We have made it clearer in the revised paper (in the caption of Figure 2).
>
> **[In section 2.3, authors described that during inference, they do not use the two matrices, but it seems difficult to understand how gamma and beta variables could be calculated using eq. 1 without them?]**
>
> During inference, we do not DIRECTLY use them for deployment since they still have large parameters. Instead, since the two matrices $ W_{c}^{\gamma} $, $W_{c}^{\beta}$ and $E^{s}$ are all fixed for each specific speaker during inference, we use the two matrices to calculate the scale and bias vectors from the speaker embedding $E^{s}$ ahead of time and use the scale and bias vectors for deployment. In this way, we can further reduce the memory storage for each speaker (as shown in Table 1 in the paper). We have improved the text description and made it clear in the revised paper.
>
> **[Describe full representation of MOS, SMOS, and CMOS]**
>
> MOS: mean opinion score. SMOS: similarity mean opinion score. CMOS: comparison mean opinion score. We have added the full representation in the revised paper.
>
> **[Other typos]**
>
> We have fixed these typos and improved the paper writing in the revised version.

---

### Official Review · AnonReviewer3 · 2020-10-28
**Interesting paper, additional comparison of results might be necessary**

**Rating:** 6
**Confidence:** 2

**Review:**

In this paper, the authors present AdaSpeech, a TTS system that can adapt to a custom voice with a high quality output and a low number of additional parameters. The model is based on the TTS model in FastSpeech 2, with several additional components. The authors show that AdaSpeech has improved results over other baselines. They also provide an interesting ablation study.

Overall, the model architecture is interesting and results seem to show its validity. However, I was wondering why the authors didn't compare their results to other known multi-speaker systems (e.g. multispeech or deepvoice 2 which were mentioned in the paper).

---

> ### Author Response · Authors · 2020-11-19
> **Response to AnonReviewer3**
>
> Thanks for your positive comments.
>
> We reply to your question on "why did not compare with other known multi-speaker systems (e.g. multispeech or deepvoice 2)" as follows.
>
> Our techniques are for high-quality and efficient TTS adaptation but not aim to model multi-speaker TTS itself, and thus are not directly comparable with other multi-speaker systems such as MultiSpeech or DeepVoice 2. However, we can apply our techniques including acoustic condition modeling and conditional layer normalization on other multi-speaker systems for TTS adaptation. We applied on MultiSpeech (which is basically a Transformer based multi-speaker system, and thus our techniques can be directly applied without much change) and compared the adaptation quality between our method and the setting that fine-tunes speaker embedding or decoder. Experiment results show that:
>
> 1. Compared with the setting that fine-tunes speaker embedding, our method achieves a CMOS score of 0.395, which shows much better adaptation quality.
> 2. Compared with the setting that fine-tunes decoder, our method achieves a CMOS score of -0.02 (usually CMOS score in the range [-0.05, +0.05] means on par quality), and greatly reduces the adaptation parameters for deployment (4.9K vs 14.1M), which is consistent with the results in our paper.
>
> These experiments demonstrate that our method is generally applicable to other multi-speaker systems.

---

### Official Review · AnonReviewer2 · 2020-10-30
**AdaSpeech review**

**Rating:** 8
**Confidence:** 5

**Review:**

This paper proposes AdaSpeech, a Transformer-based TTS architecture derived from FastSpeech, but multi-speaker, and focussed on the task of low-resource, robust, and low-dimensional speaker adaptation. The tactic for speaker modelling is that the speaker conditions only the scale and bias terms in the decoder. I don't have a clear intuition for exactly what kind of effect this would have on the phoneme embeddings and their mapping to spectral features, given that there are several non-linearities involved, but it certainly is a strong restriction. A global acoustic embedding conditions the decoder in addition to speaker embeddings, in the hopes of accounting for recording conditions, and, I suppose, timbre, which should then be disentangled from the linguistic information from the text in the decoder during pretraining and adaptable to new recordings at fine-tuning/inference. There is also a phoneme-level acoustic embedding which is used in the same way, which at inference is taken from random sentences (why not in training?), and, I guess, is supposed to cover phoneme-level idiosyncrasies of the speaker, although this isn't clear to me. However notice that, if I'm not mistaken, these acoustic embeddings are used zero-shot; it is *only* the speaker embedding that is the input to fine-tuning, and this only via the normalization parameters. (Both the normalization parameters *and* the speaker embedding itself are fine-tuned.) I am not sure what the speaker-embedding is left to do with all this acoustic-level input, but OK. The authors assert in section 2.2 that zero-shot is not enough, but they do not cite a paper that does exactly what they did. This would be useful, or, even more welcome, an ablation study with the acoustic embeddings but not the speaker embedding. Looking at Figure 4b just underscores this point for me. The result is that, within the margin of error, this method is just as good in terms of speaker similarity as fine-tuning the entire decoder. Having listened to the examples they gave, I do find that there are speakers for which it is clearly not as good, but this is not reflected in the evaluator's results.

Overall, this is very exciting work, as it not only promises space-efficient voice cloning, but, in doing so, suggests better disentanglement of speaker and phoneme properties in multi-speaker synthesis.

---

> ### Author Response · Authors · 2020-11-19
> **Response to AnonReviewer2**
>
> Thanks for your detailed and positive comments. We reply to your questions as follows.
>
> **[The authors assert in section 2.2 that zero-shot is not enough, but they do not cite a paper that does exactly what they did]**
>
> In zero-shot setting [1][2][3], they usually use a speaker encoder (not fine-tuned) to extract the characteristics of speaker from a reference audio, which is used to synthesize speech with the corresponding speaker characteristics. We have added these citations in the revised paper.
>
> [1] Neural Voice Cloning with a Few Samples, NeurIPS 2018
> [2] Transfer Learning from Speaker Verification to Multi-speaker Text-To-Speech Synthesis, NeurIPS 2018
> [3] Zero-shot multi-speaker text-to-speech with state-of-the-art neural speaker embeddings, ICASSP 2020
>
> **[This would be useful, or even more welcome, an ablation study with the acoustic embeddings but not the speaker embedding.]**
>
> We have conducted this experiment. Using the acoustic embeddings but removing the speaker embedding causes a CMOS drop of -0.249 on the VCTK datasets compared with the setting that uses both acoustic embeddings and speaker embedding, which verifies the effectiveness of speaker embedding to capture the speaker-level acoustic information.
>
> **[Having listened to the examples they gave, I do find that there are speakers for which it is clearly not as good, but this is not reflected in the evaluator's results.]**
>
> MOS is obtained by averaging the scores among all the test utterances. Since we randomly choose the demo cases, there may exist some rare cases where AdaSpeech performs close to or slightly worse than Baseline (decoder). But generally, AdaSpeech is slightly better than Baseline (decoder). We also show more demo voices by AdaSpeech and Baseline (decoder) on the demo page (https://adaspeech.github.io/#AnonReviewer2).  Note that our advantage over Baseline (decoder) is that we can greatly reduce the adaptation parameters (4.9K vs 14.1M), and achieve on par or slightly better adaptation quality.

---

### Official Review · AnonReviewer5 · 2020-11-05
**Weak Reject: Well written paper, but too applied and slightly lacking in novelty**

**Rating:** 4
**Confidence:** 5

**Review:**

### Summary

AdaSpeech is a paper on practical TTS custom voice adaptation with the aim of reducing the amount of adapted parameters per voice to allow cloud serving of a large number of custom voices while maintaining high adaptation quality and similarity. The novel piece that enables this is the conditioning of layernorm in the model on the speaker embedding. The grammar reads slightly awkwardly in places, but the paper is understandable and well structured. Descriptions of the model, experiments, and analysis of results are well done.

### Recommendation

**Weak Reject**

I believe this paper is not novel enough / too applied / focused on experimental results for this conference. There is little discussion on the theoretical side of the acoustic condition modelling, such as how the authors are able to determine that the utterance-level and phoneme-level vectors are modelling things like room condition. Instead, the strengths of this paper are entirely through the strong numerical results. I think this paper would be a solid accept for a more specialized conference like ICASSP or Interspeech.

### Positives

1. Well written, great analysis of results and ablation studies.

### Negatives

1. What is the loss used to train the phoneme level acoustic predictor? MSE?

1. How is it determined that acoustic conditions such as loudness or room conditions are actually captured by the utterance- and phoneme-level acoustic condition modelling? My intuition would be that your phoneme-level predictor is trained only with phoneme hiddens (textual information only) (do these phoneme hiddens include speaker embedding information?), so at most it models some pitch or prosody information. The utterance-level can definitely model the rest, but where is the evidence? It could end up modelling only one very specific dimension and still improve the MOS.

1. Similarly, I highly doubt the utterance-level acoustic condition modelling does not also capture speaker information. What happens when using a speaker embedding with the utterance-level vector extracted from a reference speech for a different speaker?

1. The paper would be better with a discussion on controllability. As a reference utterance needs to be provided, does this mean the synthesized speech will take on the prosody in the reference? What happens if you want to synthesize a prosody for a speaker that's not present in any of that speaker's reference utterances? I understand this is a big topic with ongoing research which is why it would be a big bonus if this paper can make any kind of progress in that area.

1. I am curious how your phoneme-level predictor would compare with a VAE-based setup, although I understand this can be difficult to set up so no action is required here.

### Misc

**2.3 Pipeline of AdaSpeech**: "we do not use the two matrices in each conditional layer normalization" -- which two matrices?

**4.2 Method Analysis**: What does it mean to remove conditional layer normalization? Then you don't have any adaptation parameters, so is it not equivalent to Baseline (spk emb) case?

---

> ### Author Response · Authors · 2020-11-19
> **Response to AnonReviewer5  (Part 1)**
>
> Thanks for your comments on our paper. We reply to your questions as follows.
>
> **[About the novelty]**
>
> We clarify the novelty of our work as follows:
> 1. We leverage acoustic condition modeling (utterance-level and phoneme-level) to better model the acoustic conditions and improve the generalizability and adaptability of the TTS model, in order to support diverse customers with different acoustic conditions for TTS adaptation.
> 2. We introduce conditional layer normalization to adapt the TTS model with high voice quality using as few adaptation parameters as possible, in order to support a large number of customers with small memory storage.
>
> The problems we solve are very critical in TTS adaptation for custom voice, and the techniques we proposed handle these problems effectively and achieve good experiment results.
>
> Besides, we also point out that some previous works assume the source speech data and adaptation data are in the same domain, which is not practical in custom voice scenarios. Our method works well for this challenging scenario: the adaptation data and source data are in different domains. The experiment results in Table 1 in the paper also echo this point:
> 1.  When the source and target domain are the same (both are LibriTTS datasets), the adaptation quality of Baseline (spk emb) are very close to the ground-truth audio.
> 2. When the source and target domain are different (pre-trained on LibriTTS but adapted on VCTK and LJSpeech), the adaptation quality of Baseline (spk emb) has large gap to the ground-truth audio. In this scenario, AdaSpeech achieves much better quality compared with Baseline (spk emb), and greatly reduces the parameters in inference by 14.1M/4.9K=2878 times compared with Baseline (decoder).
>
> **[The fitness of our paper to machine learning conference]**
>
> We want to point out that:
> 1. Many previous application-oriented papers on TTS (including TTS adaptation), e.g., [1,2,3,4,5,6,7,8], have been published in a variety of machine learning conferences, including ICLR, ICML and NeurIPS.
> 2. Many algorithmic/applied papers without theoretical analysis are published in ICLR, e.g., [9,10].
> 3. Many papers with only experimental studies are published in ICLR, e.g., [11,12]. These are all good works.
>
> Our work solves an important problem in TTS adaptation and has good method formulation and solid experiment results, which we think well fits the standard of ICLR 2021 conference.
>
> [1] Deep Voice: Real-time Neural Text-to-Speech, ICML 2017
> [2] Deep Voice 2: Multi-Speaker Neural Text-to-Speech, NIPS 2017
> [3] Deep Voice 3: Scaling Text-to-Speech with Convolutional Sequence Learning, ICLR 2018
> [4] Towards End-to-End Prosody Transfer for Expressive Speech Synthesis with Tacotron, ICML 2018
> [5] Neural Voice Cloning with a Few Samples, NeurIPS 2018
> [6] Transfer Learning from Speaker Verification to Multi-speaker Text-To-Speech Synthesis, NeurIPS 2018
> [7] Sample Efficient Adaptive Text-to-Speech, ICLR 2019
> [8] FastSpeech: Fast, Robust and Controllable Text to Speech, NeurIPS 2019
> [9] A Universal Music Translation Network, ICLR 2019
> [10] BERTScore: Evaluating Text Generation with BERT, ICLR 2020
> [11] An Empirical Study of Example Forgetting during Deep Neural Network Learning, ICLR 2019
> [12] Cross-Lingual Ability of Multilingual BERT: An Empirical Study, ICLR 2020

---

> > ### Author Response · Authors · 2020-11-19
> > **Response to AnonReviewer5 (Part 2)**
> >
> > **[Discussion on how the acoustic condition modeling works]**
> >
> > 1. Why use acoustic condition modeling?
> >     - It is challenging for TTS adaptation since the source speech cannot cover all the acoustic conditions in custom voice. A practical way to alleviate this issue is to improve the adaptability (generalizability) of source TTS model. However, in text to speech, since the input text lacks enough acoustic conditions (such as speaker timbre, prosody and recording environments) to predict the target speech (text-to-speech mapping is a one-to-many mapping problem), the model tends to memorize and overfit on the training data and has poor generalization during adaptation.
> >     - To solve this problem, we use acoustic condition modeling to provide corresponding acoustic conditions (e.g., speaker timbre, prosody and recording conditions) as input in training to make the model learn reasonable one-to-one text-to-speech mapping towards better generalization instead of memorization.
> >
> > 2. How can the acoustic condition modeling help?
> >     - In training, we use phoneme-level and utterance-level acoustic encoders to extract necessary acoustic conditions from the reference (target) speech in order to help the decoder to learn to predict the corresponding mel-spectrogram instead of learning to memorize.
> >     - The utterance-level vector is a summary of the whole utterance, which tends to model general speaker information, room condition, etc.
> >     - The phoneme-level vectors are extracted from the averaged speech sequence (averaged from frame level into phoneme level) and are of low dimension (only 4 as shown in the Model Configurations part in Section 3 of the paper), which tend to model fine-grained information like prosody, pitch, etc., instead of general speaker or room condition information.
> >     - We conducted preliminary studies on the dimension of the phoneme-level vectors, and found that larger dimensions result in worse adaptation quality and thus low dimension is important to only extract information like prosody.
> >     - Therefore, in inference, phoneme-level predictor can predict prosody information based on phoneme hiddens.
> >
> > 3. How to determine to model acoustic conditions like room condition?
> >     - We do not aim to precisely determine to model which types of condition information, since it depends on what kind of acoustic information the speech has. We just let the two acoustic encoders to extract necessary information from different granularities to predict the correct mel-spectrogram, instead of overfitting due to lack of enough acoustic information.
> >     - By designing the utterance encoder to extract a single vector and the phoneme encoder to extract a sequence of low-dimension vectors, they can be automatically leant to represent speaker or room information in utterance vector, and prosody information in phoneme vectors. If the target speech has room condition noises, the acoustic encoder can learn to extract this kind of information to make correct prediction.
> >
> >
> > **[What is the loss used to train the phoneme-level acoustic predictor? MSE?]**
> >
> > Yes, we use MSE to train the phoneme-level acoustic predictor, where the label is the sequence of vectors generated by the well-trained phoneme-level acoustic encoder.
> >
> >
> > **[Using a speaker embedding with the utterance-level vector extracted from a reference speech for a different speaker]**
> >
> > The utterance-level acoustic encoder tends to extract utterance-level acoustic information, such as speaker information and room condition in this utterance. When using a speaker embedding with the utterance-level vector extracted from a reference speech for a different speaker, the synthesized speech will have mixed speaker information (including the information from the speaker embedding of one speaker and that from the utterance-level vector of another speaker). We also show some demo voices on the demo page (https://adaspeech.github.io/#AnonReviewer5).

---

> > > ### Author Response · Authors · 2020-11-19
> > > **Response to AnonReviewer5 (Part 3)**
> > >
> > > **[The paper would be better with a discussion on controllability]**
> > >
> > > 1. As we clarify in [Discussion on how the acoustic condition modeling works] in our previous response (part 2), our acoustic condition modeling aims to provide necessary acoustic information for speech prediction, which can make the TTS model more generalizable (adaptable) to support diverse custom voices, instead of memorizing due to lack of acoustic information in prediction. Therefore, we focus more on generalizability and adaptability, instead of controllability.
> > > 2. Controllability is another important and interesting topic for TTS. While this paper mainly achieves high-quality and efficient adaptation on custom voice, it has potential to control the synthesized voice. For example, we can choose different reference speech to synthesize speech in a similar timbre or style with this reference speech. We can also add a VAE or VQ-VAE module in the utterance-level and phoneme-level acoustic modeling, where the latent vector or latent ID can be adjusted to synthesize speech with different prosodies. We leave controllability for future work.
> > >
> > > **[I am curious how your phoneme-level predictor would compare with a VAE-based setup]**
> > >
> > > Actually, we conducted the VQ-VAE experiment before paper submission, where we quantize the sequence of vectors generated by the phoneme-level acoustic encoder into a sequence of codebook ID, each ID is associated with an embedding vector and is then taken as input to the decoder. VQ-VAE achieves slightly worse voice quality (a CMOS score of -0.138) compared with our default setting. It seems that making the hidden vector discretized harms the voice quality a little. We will explore more on the controllability of the model and try VAE in the future.
> > >
> > > **[We do not use the two matrices in each conditional layer normalization" -- which two matrices?]**
> > >
> > > The two matrices refer to the $W_{c}^{\gamma}$ and $W_{c}^{\beta}$ in each conditional layer normalization in decoder, as shown in Equation 1 in the paper. During inference, we do not directly use them for deployment since they still have large parameters. Instead, since the two matrices $W_{c}^{\gamma}$ and $W_{c}^{\beta}$, and speaker embedding $E^{s}$ are all fixed for each specific speaker during inference, we calculate the scale and bias vectors according to the two matrices and speaker embedding ahead of time and use the scale and bias vectors for deployment. In this way, we can further reduce the memory storage for each speaker (as shown in Table 1 in the paper). We have improved the text description and made it clear in the revised paper.
> > >
> > >
> > > **[What does it mean to remove conditional layer normalization?]**
> > >
> > > When removing conditional layer normalization, it degenerates to conventional layer normalization without conditioning on speaker embedding. In this way, the speaker embedding is still added into the hidden sequence and taken as the input to the decoder, as shown in Figure 2(a) in the paper. We only fine-tune the speaker embedding in this setting. This setting is not equivalent to Baseline (spk emb), but equivalent to Baseline (spk emb) + utterance-level and phoneme-level acoustic condition modeling.

---

> > ### Comment · AnonReviewer5 · 2020-11-24
> > **Re: Response to AnonReviewer5**
> >
> > Thank you for the detailed response. I will retract my statements regarding suitability for the conference as I am definitely not clear on how to judge that. I will provide a score without considering that.
> >
> > **[About the novelty]**
> >
> > _1. We leverage acoustic condition modeling (utterance-level and phoneme-level) to better model the acoustic conditions and improve the generalizability and adaptability of the TTS model, in order to support diverse customers with different acoustic conditions for TTS adaptation._
> >
> > In my understanding, the separate modelling of utterance-level and phoneme-level acoustic condition has already been demonstrated by _Sun et al._, **Fully-hierarchical fine-grained prosody modeling for interpretable speech synthesis**, ICASSP 2020, in Section 3:  _The model incorporates the utterance-level feature $z^u$ by conditioning other fine-grained latent features on the utterance level latent one, and subtracting [...] in Eq.(1) accordingly._ I do not see how the approach proposed in this work is fundamentally different. In fact, it seems to be weaker, as there is no constraint preventing the phoneme-level and utterance-level acoustic encoders from modelling the same acoustic conditions.
> >
> > The use of a separate coarse and fine latents for few-shot adaptation has also been explored in _Choi et al._, **Attentron: Few-Shot Text-to-Speech Utilizing Attention-Based Variable-Length Embedding**, INTERSPEECH 2020, although that work is considered under "very recent work" for this submission.
> >
> > As such, from what I can see, the novel contributions from this work is just the conditional layer normalization significantly reducing the number of parameters that need to be adapted. I do think this is an excellent contribution, but still I do not think it is enough.
> >
> > **[Discussion on how the acoustic condition modeling works]**
> >
> > _The utterance-level vector is a summary of the whole utterance, which tends to model general speaker information, room condition, etc. The phoneme-level vectors [...] tend to model fine-grained information like prosody, pitch, etc., instead of general speaker or room condition information. By designing the utterance encoder to extract a single vector and the phoneme encoder to extract a sequence of low-dimension vectors, they can be automatically leant to represent speaker or room information in utterance vector, and prosody information in phoneme vectors. If the target speech has room condition noises, the acoustic encoder can learn to extract this kind of information to make correct prediction._
> >
> > I agree that this is what theory would say, but I wanted to see experiments verifying this in the paper. For example, show that using a utterance-level reference with room condition noises results in the same room condition noises in the output, but phoneme-level vectors extracted from the same noisy reference does not result in noises in the output. Instead, there is no evidence for any of these claims other than a general improvement in MOS which could be due to various confounding factors.
> >
> > **[Yes, we use MSE to train the phoneme-level acoustic predictor]**
> >
> > Please mention this in the paper. I cannot seem to find where it is mentioned.
> >
> > **[Using a speaker embedding with the utterance-level vector extracted from a reference speech for a different speaker]**
> >
> > Thank you for providing samples with this experiment. I think it is unfortunate, although expected, that the utterance-level vector is entangled with the speaker identity. What this means is that in the custom voice inference case, the utterance-level vector must be extracted from a reference speech of that custom voice, so in the case where there is few examples of the custom voice, the number of possible output styles from this model will be limited. For example, if the speaker only records 1 minute of flat monotone speech, it will not be possible to generate speech for that speaker with wild prosody.
> >
> > While such a result may not have been a goal of this work, it would have definitely increased my rating significantly.
> >
> > **Taking the above into consideration, I would like to keep my rating at 4.**

---

> > > ### Author Response · Authors · 2020-11-24
> > > **The 2nd response to AnonReviewer5 (Part 1)**
> > >
> > > We appreciate your active discussions on our first response. Here is our second response to your detailed comments.
> > >
> > > **[About the novelty]**
> > >
> > > 1. _The separate modelling of utterance-level and phoneme-level acoustic condition has been used by previous works._
> > >
> > >     - We did not claim that we are the first to propose utterance- and phoneme-level modeling in TTS.
> > >         - Actually, we have already discussed the differences with previous works in the submitted version (the last paragraph of Section 2.1). Previous works usually use speech encoders to extract a single vector and/or a sequence of vectors to represent the characteristics of a speech sequence to improve the speaker timbre or prosody, or improve the controllability of the model. The paper ‘Fully-hierarchical fine-grained prosody modeling for interpretable speech synthesis’ you mentioned also focuses on the disentangled control of prosody.
> > >         - We do not focus on improving the prosody or controllability of multi-speaker model itself. Instead, the highlight of our acoustic condition modeling is the novel perspective to model the diverse acoustic conditions in different granularities to provide necessary input information, in order to make the source TTS model more adaptable to different environments and acoustic conditions. Experiments demonstrate the success and effectiveness of this technique in the challenging TTS adaptation for custom voice, where both model generalizability (adaptability) and efficient adaptation with few parameters are critical.
> > >
> > >     - On the other hand, the novel application of existing techniques and ideas is highly appreciated by the machine learning community. For example, applying Transformer into different tasks including image generation [1], music generation [2] and image classification [3], applying the idea of contrastive learning into different domains including image [4], reinforcement learning [5] and speech [6]. We leverage utterance-level and phoneme-level modeling to improve the generalizability (adaptability) of TTS model for the challenging TTS task of custom voice, which is a novel application of the techniques.
> > >     [1] Image Transformer, ICML 2018
> > >     [2] Music Transformer: Generating Music with Long-Term Structure, ICLR 2019
> > >     [3] An Image is Worth 16x16 Words: Transformers for Image Recognition at Scale, ICLR 2021 submission (good rating score)
> > >     [4] ContraGAN: Contrastive Learning for Conditional Image Generation, NeurIPS 2020
> > >     [5] CURL: Contrastive Unsupervised Representations for Reinforcement Learning, ICML 2020
> > >     [6] wav2vec 2.0: A Framework for Self-Supervised Learning of Speech Representations, ICML 2020
> > >
> > >
> > > 2. _It seems to be weaker, as there is no constraint preventing the phoneme- and utterance-level acoustic encoders ..._
> > >
> > >   Since the goal of this work is not disentanglement/controllability but to improve the generalizability/adaptability of the source TTS model, we do not add explicit constraints on what the phoneme- and utterance-level acoustic encoders should learn, as long as they can learn the necessary information to help the model generalize. Instead, by designing the utterance encoder to extract a single vector and the phoneme encoder to extract a sequence of low-dimension vectors, they implicitly learn different acoustic information.
> > >
> > > 3. _the "very recent work" in INTERSPEECH 2020_
> > >
> > >     As you mentioned, it is a very recent work. Besides, our method is different from theirs since they use multiple reference utterances via the attention mechanism. Moreover, in their adaptation setting, VCTK datasets are used both in the source (pre-training) and target (adaptation) domain, which is much simpler, as we stated in the last rebuttal (“Response to AnonReviewer5 (Part 1)” [About the novelty]). We mainly consider the challenging setting in custom voice with target domains different from the source domain. Last but not the least, the VCTK datasets contain a large amount of overlapping text between different speakers (since VCTK is originally designed for voice conversion, and you can refer to the VCTK link https://datashare.is.ed.ac.uk/handle/10283/3443), which greatly simplifies the adaptation setting in those papers that use VCTK as both the source and target domains, since the text used in adaptation and inference is likely to be already seen in the training of the source TTS model.
> > >
> > > 4. _Summary_
> > >
> > >   Overall, our contributions include 1) the novel perspective to leverage phoneme- and utterance-level modeling to improve the generalizability/adaptability of source TTS model; 2) introducing conditional layer normalization to greatly reduce the adaptation parameters but maintain high voice quality; and 3) addressing the challenging and realistic adaptation setting (source/target domains are different) in custom voice (which is not considered by most previous TTS adaptation works). We think our contributions are strong enough for high-quality conferences like ICLR.

---

> > > > ### Author Response · Authors · 2020-11-24
> > > > **The 2nd response to AnonReviewer5 (Part 2)**
> > > >
> > > > **[Discussion on how the acoustic condition modeling works——show the voice]**
> > > >
> > > > Thanks for your advice. We are preparing the voice now and will update to you soon when it is ready.
> > > >
> > > > **[Yes, we use MSE to train the phoneme-level acoustic predictor——mention in the paper]**
> > > >
> > > > Thanks for your suggestion. We have added in the ‘Training, Adaptation and Inference’ part in Section 3 of the revised paper.
> > > >
> > > > **[Using a speaker embedding with the utterance-level vector extracted from a reference speech for a different speaker——flat prosody]**
> > > >
> > > >   - Thanks for your advice. Disentangling speaker or other acoustic information, or improving the controllability and style in inference are interesting and ongoing topics in general TTS research.
> > > >   - However, they are not our focus in this paper. Instead, we focus on 1) improving the generalizability and adaptability of the TTS model in order to support diverse customers; 2) adapting the TTS model with high voice quality using as few adaptation parameters as possible in order to support a large number of customers with small memory storage. Modeling the acoustic information (e.g., speaker, prosody, room conditions) is mainly for improving the generalization of TTS model, instead of for prosody-rich speech synthesis. Actually, we only use 20 sentences (about 1 minute speech) for adaptation in our experiments, and results in Table 1 show good MOS and SMOS scores.
> > > >   - Anyway, controllable, expressive, and prosody-rich speech synthesis is definitely our future research focus. We will continue to extend AdaSpeech and further explore in this area in the future.
> > > >
> > > >
> > > > **Hopefully our response can clarify these points and help you better recognize the value of our work. Thanks very much for your time and effort for the detailed comments!**

---

> > > > ### Author Response · Authors · 2020-11-25
> > > > **The 2nd response to AnonReviewer5 (Part 2 Update)**
> > > >
> > > > **Update for the previous response: The 2nd response to AnonReviewer5 (Part 2)**
> > > >
> > > > **[Discussion on how the acoustic condition modeling works——show the voice]**
> > > >
> > > >   We conducted several experiments to analyze the acoustic condition modeling. We describe the results as follows.
> > > >   - In the first experiment, we only use clean adaptation data to adapt a speaker. In inference, we use an utterance-level vector extracted from a noisy reference speech to synthesize speech. The voices can be found here https://adaspeech.github.io/#AnonReviewer5exp1. We show two cases, each case in one line. The first audio in each line is the synthesized speech with noisy reference speech (the second audio). The third audio in each line is the synthesized speech with clean reference speech (the fourth audio). We can see that the synthesized speech is noisy when the reference speech is noisy. It shows the utterance-level vector indeed contains noise information, which causes the synthesized speech to be noisy.
> > > >   - In the second experiment, we only use noisy adaptation data to adapt a speaker. In this way, the reference speech for the utterance-level and phoneme-level encoders during adaptation (fine-tuning) is noisy. In inference, we use an utterance-level vector extracted from a clean reference speech to synthesize speech. The voices can be found here https://adaspeech.github.io/#AnonReviewer5exp2. In the first line, we show some noisy samples that are used for adaptation. The second and third lines show the synthesized speech with the corresponding clean reference speech, respectively. We can see that the synthesized speech is clean, which shows that the phoneme-level vectors predicted by the phoneme-level acoustic predictor do not contain noisy information.
> > > >   - To better demonstrate the advantages of phoneme-level acoustic modeling in improving the prosody, we show some demo voices generated by our AdaSpeech with and without phoneme-level acoustic modeling in https://adaspeech.github.io/#AnonReviewer5exp3. You can hear that phoneme-level acoustic modeling can indeed improve the prosody of the synthesized speech.
> > > >
> > > >
> > > > **Hopefully our response can clarify these points and help you better recognize the value of our work. Thanks very much for your time and effort for the detailed comments!**

---

### Decision · Program_Chairs · 2021-01-07
**Final Decision**

**Decision:**

Accept (Poster)

**Comment:**

The paper is about adapting a voice generation model to new speakers with minimal amount of training data. The key insight in this paper is that the voice can be adapted using a small set of variables -- the bias and the variance associated with the layer that normalizes the mel-spectrogram associated with the decoder. Additionally, they characterize voice at the utterance level to capture stationary factors like background acoustic conditions and at the phoneme level to capture factors such as prosody, though there are no explicit constraints to force such representation.

The strength of the paper are:
+ Simplicity of the approach
+ Empirical evaluation that demonstrates its effectiveness

The weakness of the paper are:
- analysis of what the crucial parameters of the model represent
- lack of clarity that is obvious from several back-and-forths between the reviewers and the author.

A few examples include:
- “There is also a phoneme-level acoustic embedding which is used in the same way, which at inference is taken from random sentences (why not in training?), and, I guess, is supposed to cover phoneme-level idiosyncrasies of the speaker, although this isn't clear to me.”
- “ it is only the speaker embedding that is the input to fine-tuning, and this only via the normalization parameters. (Both the normalization parameters and the speaker embedding itself are fine-tuned.) I am not sure what the speaker-embedding is left to do with all this acoustic-level input, but OK.”